# Anti-Inflammatory Benefits of Food Ingredients in Periodontal Diseases

**DOI:** 10.3390/pathogens12040520

**Published:** 2023-03-27

**Authors:** Evangelos Papathanasiou, Reem Alreshaid, Mariely Araujo de Godoi

**Affiliations:** 1Department of Periodontology, Tufts University School of Dental Medicine, Boston, MA 02111, USA; 2Department of Diagnosis and Surgery, School of Dentistry at Araraquara, UNESP, Araraquara 14801-385, SP, Brazil

**Keywords:** supplements, inflammation, periodontitis, gingivitis, immune response, polyphenols, vitamins, omega-3

## Abstract

Periodontitis is a multi-faceted inflammatory disease that impacts the gingiva and the structures that support our teeth, and may eventually increase tooth mobility and the risk of tooth loss. Inflammation is a viable therapeutic target of periodontitis for both biologic (dietary) and host modulatory agents/drugs. Conventional therapeutic approaches for periodontitis, including nonsurgical or surgical periodontal therapy as well as occasional adjunctive antimicrobial therapy, have been only marginally effective. Malnutrition, or at least poor dietary habits, can be highly prevalent among patients with periodontal diseases. As several food nutrients can aid in periodontal healing and regeneration, there is a critical need to evaluate natural dietary sources and supplement ingredients that can counterbalance the inflammatory processes and improve the periodontal status of our patients. Here, we reviewed the current state of knowledge (search period: 2010 to 2022; PubMed and Web of Science) on the anti-inflammatory actions of food ingredients and supplements in clinical studies of patients with periodontal diseases. A diet that includes fruits and vegetables, omega-3 polyunsaturated fatty acids, and supplements of vitamins and plant-derived compounds seems to counteract gingival inflammation and has a promising therapeutic impact in patients with periodontal diseases. Despite the positive indications that several nutrients can be used as an adjunct to periodontal therapy, additional studies with bigger sample sizes and longer follow-up periods are needed to elucidate their therapeutic benefits and the most effective doses and administration.

## 1. Introduction

Periodontitis is the sixth most common health condition in the world and the primary reason for tooth loss in adults [1]. At least 47% of adults aged 30 and older in the United States have periodontitis, among whom 7.8% are diagnosed with severe periodontitis [2]. Periodontitis is characterized by the inflammation of the gingiva and the surrounding structures of the teeth, which is triggered by dental bacterial plaque. If periodontitis is not treated, destruction of the periodontal attachment apparatus may occur, which can increase tooth mobility and the risk of tooth loss [3].

Periodontitis is a multifaceted chronic disease involving constant interactions between bacteria and the host immune/inflammatory response, which are modified by genetic and environmental factors. Although dental bacterial plaque has been mainly considered the crucial triggering factor for periodontal diseases, patients do not all have the same susceptibility and the same response to treatment; the host immune response is ultimately what largely guides the pathological process [4]. Periodontal health is characterized by stable periodontal microflora and a controlled host immune/inflammatory response which coexist in a state of dynamic equilibrium, known as symbiosis or homeostasis. Dysbiosis can disrupt the homeostasis in periodontal tissues by either dysregulating the inflammatory response or stimulating an imbalance within the microbial community, which ultimately leads to a hyperinflammatory immune response with a major release of IL-1 superfamily members [5], extracellular matrix degradation, and alveolar bone loss [6]. The severity of periodontal tissue breakdown is finally determined by endogenous mechanisms that regulate the inflammatory mediators and the balance between pro- and anti-inflammatory mediators [7]. 

Dental cleaning, subgingival scaling/root planning (SRP), and meticulous oral hygiene are the main traditional therapeutic approaches for treating periodontal diseases. These approaches attempt to reduce the levels of pathogenic bacteria in periodontal pockets [8]. Our standard of care remains nonsurgical periodontal therapy, which mechanically eliminates the etiologic dental biofilm, whether antimicrobials are combined or not. However, targeting only bacteria does not always allow us to achieve a desirable outcome in every patients with periodontitis [9]. While this therapeutic approach is successful to some extent, the lack of control of the host inflammatory response remains a threat and merits more attention. 

Inflammation is a viable therapeutic target of periodontitis, for both biologic (dietary) and host modulatory agents/drugs [10]. Periodontal disease progression can be prevented by the adjunctive use of host modulatory agents, especially in patients with a higher risk to present with a chronic (hyper) inflammatory response to the bacterial plaque, which is linked to genetic, systemic, or environmental factors, and in patients for whom traditional therapeutic methods are not successful [10]. Malnutrition, or at least poor dietary habits, can be highly prevalent among patients with periodontal diseases. The intake of food supplements has become very popular, as they can improve our health, support our immune system, and reduce the risk of several chronic diseases. As several food nutrients can aid in periodontal healing and regeneration, there is a critical need to evaluate natural dietary sources and supplement ingredients that can counterbalance the inflammatory processes and improve the periodontal condition of our patients. The majority of the previous review papers in this field have combined data from in vitro, in vivo, and human clinical studies. The aim of this paper is to provide a critical review of the most updated evidence focused solely on human clinical studies on food ingredients and supplements (oral intake) that exhibit anti-inflammatory benefits against periodontal disease.

## 2. Materials and Methods

An electronic database search on MEDLINE (PubMed) and Web of Science was conducted to identify empirical human clinical studies on the anti-inflammatory effects of different nutrients on periodontal diseases published between 1 January 2010 and 31 December 2022. The electronic search using the combination of keywords that represent periodontal diseases (Periodontal OR Periodontitis OR Gingivitis) and those that represent the nutrients (Supplements OR Diet OR Nutrition, OR Food OR Omega OR Probiotics OR Vitamins OR Polyphenols OR Natural products) allowed us to identify 553 possible eligible studies. After excluding duplicate articles, the number of studies decreased to 303. The 303 articles from the electronic search were further filtered by selecting only randomized clinical trials, case-control studies, and cohort studies published in English, as well as studies conducted only in human subjects (clinical studies). We further manually searched all available studies and excluded any formulations of different nutrients in oral health products (mouthwash, toothpaste, gels, strips) that could be used for the management of periodontal diseases, as we would like to focus our review only on the oral intake of nutrients. The total number of articles that were included in this review were 46 clinical studies, which are summarized into different nutrient categories and presented in tables throughout the manuscript.

### 2.1. Diet

A well-balanced daily diet plays a crucial role in our general health and in supporting our defense against infections, including those from periodontal pathogens and bacterial plaque. 

Nutrients derived from our diet are necessary for our health as they provide a vital energy source (macronutrients) as well as essential cofactors required for enzyme function, structural elements, and transport (micronutrients) [11]. Studies on patients with a Western-type diet (WD), which characterizes the diet in Western societies and includes high amounts of refined grains, sugar, and saturated, trans, and omega-6 fatty acids and low amounts of micronutrients and fibers, favored a pronounced gingival inflammatory response [12,13]. On the other side, a Mediterranean diet, which mainly characterizes the diet of southern Italians and Greeks around the 1950s and 1960s, is considered the “gold standard” diet for maintaining optimal health, with benefits in weight loss and in decreasing the risk of several chronic conditions, including diabetes mellitus and metabolic syndrome [13]. The modification of our diet can also reduce periodontal inflammation and the risk of periodontal diseases, according to studies described below and summarized in Table 1.

A recent randomized controlled trial (RCT) evaluated the impact of the Mediterranean diet on gingival inflammation [13]. The experimental group of subjects had to modify their diets for 6 weeks into one that included a high intake of vegetables, fruits, olive oil, herbs, and fatty sea fish, while a control group did not change their habitual diet. While dental bacterial plaque scores remained similar between the groups, gingival inflammation was reduced, with decreased gingival index (GI) and bleeding on probing (BOP) scores in subjects who adhered to the Mediterranean diet [13]. This RCT confirmed findings from previous pilot studies showing that switching to a diet low in carbohydrates, high in omega-3 fats, and rich in vitamins C and D, antioxidants, and fiber for 4 weeks results in a decrease in all periodontal inflammatory parameters, to approximately half of the baseline values [14,15]. Another pilot study in patients with gingivitis showed that treatment with an eight-week high-fiber, low-fat diet significantly improved markers of periodontal disease, as well as metabolic profiles (body weight, HbA1c, and high C-reactive protein levels) [16]. 

A large population-based survey in Finland (*n* = 2187 subjects) using a validated food frequency questionnaire showed that the Nordic diet, which is rich in fruits, vegetables, cereals, and fish and low in unsaturated fats, was inversely associated with gingival bleeding and periodontal pocket depth (PPD) in patients with poor oral hygiene [17]. A recent population-based study in Malmö, Sweden, showed that fatty fish consumption was associated with reduced gingival inflammation and periodontal pockets [23]. A diet rich in nitrate by repeated lettuce juice consumption in gingivitis patients [18] and patients with a history of periodontitis (reduced periodontium) [19] resulted in a reduction in gingival inflammation and bleeding on probing (BOP), accompanied by significant compositional changes within the subgingival microbiome 14 days after periodontal recall care. Although the Atkins low-carb ketogenic diet has been shown to reduce the risk of inflammatory diseases, it did not improve clinical periodontal parameters in a 6-week period [20]. 

Fruit is often endorsed as a nutrient source rich in vitamins, antioxidants, and fiber, and its consumption is reported to counteract inflammation [24]. Diets rich in fruit and vegetables have been positively associated with periodontal health and with improved healing after periodontal therapy [25]. In a recent RCT, 30 healthy, non-obese participants were randomly distributed to either supplement their diet with extra fruits or nuts, each at +7 kcal/kg body weight/day, for two months. A large increase in fruit intake resulted in a significant reduction in the number of subjects with PPD ≥4 mm, compared to an increased intake of nuts [21]. The consumption of two kiwifruits/day for two months reduced gingival bleeding in patients with periodontitis without any other periodontal treatment [22]. Interesting findings were also obtained from another RCT on gingivitis patients, where the consumption of a cranberry functional beverage as an adjunct to nonsurgical periodontal therapy resulted in a significant reduction in gingival index (GI) and plaque index (PI), but not BOP, compared to patients who consumed only water [26]. 

In our next section, we describe how specific micronutrients and vitamins in the diet counteract with gingival inflammation and explain the contribution of a diet rich in fruit and vegetables on periodontal health. 

### 2.2. Vitamins

Vitamins play a key role in the coordination of the immune system and in the prevention and delay of the progression of several diseases. Vitamins have already been approved to have anti-inflammatory and antibacterial effects, in addition to regulating osteoblastic function in periodontal diseases, which makes them great targets as adjunctive treatments in patients with periodontitis [27]. The impact of vitamin deficiencies on periodontal diseases has been reviewed thoroughly before [11,28], so here, we focus on human clinical studies on vitamin supplementation as adjuncts to periodontal treatment, which are described in the next section and summarized in Table 2. 

Vitamin D stands out as the main target of research studies in periodontal therapy. 360 patients with moderate or severe periodontitis were randomly prescribed, after 3 months of nonsurgical periodontal therapy, 2000 international units (IU)/d of vitamin D3, 1000 IU/d of vitamin D3, or placebo for 3 months [29]. Vitamin D supplementation in both dose groups resulted in a slight, but significant, reduction in PPD and clinical attachment loss (CAL) for moderate and deep periodontal pockets. However, it was concluded that vitamin D supplementation had a modest effect as an adjunct to periodontal therapy, with limited clinical relevance. A possible limitation of that study [29] could be the short follow-up period (3 months). Another RCT with a significantly smaller sample of patients with periodontitis but for a longer period of evaluation, 6 months, evaluated the effectiveness of a higher dose of vitamin D supplementation (25,000 IU/week; ~3570 IU/day)) with nonsurgical periodontal treatment. Both the experimental and control groups showed improvements in all evaluated clinical parameters (PPD, BOP, PI); however, the reduction in PPD was greater in the test group that received the supplementation of vitamin D [30]. In another RCT, a dose of 400 IU/day of vitamin D supplementation for 6 weeks as an adjunct to nonsurgical periodontal therapy resulted in similar improvements in periodontal indices compared to nonsurgical periodontal treatment [31]. In this study, the authors also evaluated bone mineral density by a qualitative ultrasound; however, no significant difference was observed between comparison groups. Very few clinical studies have investigated changes in subclinical markers such as reductions in gingival crevicular fluid (GCF) and serum IL-1β [32] and decreases in serum cytotoxic T cells (CD3 and CD8) and of salivary cytokines [33] associated with vitamin D supplementation. The crucial role of vitamin D in maintaining periodontal health has been recognized; however, more clinical trials with longer follow-up periods are needed to determine the most efficient dosage and duration of vitamin D supplementation. 

Vitamin C deficiency has been strongly associated with severe gingivitis, periodontal disease, and generalized gingival bleeding. In a case-control study, periodontitis patients were shown to have lower plasma vitamin C levels than individuals without periodontal breakdown [42]. In another case-control study, periodontitis patients with tooth loss reported a higher intake of vitamin C compared to patients without periodontitis [35]. The benefits of combining vitamin C supplements with periodontal therapy have not been investigated enough with clinical studies. Only one RCT evaluated the supplementation of vitamin C with nonsurgical periodontal therapy for 4 weeks [34], without showing any additional improvement in periodontal parameters in the test group. However, in this RCT, only 20% of the periodontitis patients were vitamin-C-depleted (plasma concentration < 4 mg/L), highlighting the importance of monitoring serum vitamin levels before adding supplements to our patients, as we do not know if they are already sufficiently replete with this added micronutrient. 

Vitamin E has important antioxidant and anti-inflammatory properties. Deficiencies in vitamin E have controversial associations with periodontitis [11]. The administration of 200 mg vitamin E every other day for 3 months in addition to SRP in periodontitis patients resulted in a significant improvement in periodontal parameters and salivary superoxide dismutase compared to SRP alone [36]. In a multicenter RCT in patients with mild to moderate periodontitis, a fixed dose of the combination of vitamin C, vitamin E, lysozyme, and carbazochrome (CELC), combined with SRP, was tested for 4 weeks and resulted in a significant reduction in GI compared to SRP and placebo, without any differences in other periodontal parameters between studied groups [37]. The adjunctive intake of 400 mcg folic acid tablets with SRP resulted in a significant additional clinical attachment level (CAL) gain in periodontitis patients compared to SRP and placebo tablets, without any significant change in biochemical parameters between the studied groups [38]. Further clinical and biochemical data in larger clinical trials are needed to support the present findings.

Combined micronutrient supplements have also been investigated in patients with periodontal diseases. Patients with severe periodontitis who adhered to the Mediterranean died and received SRP with multi-micronutrient food supplement administration, including the co-enzyme Q10 multicomposite, docosahexaenoic acid (DHA), *B. serrata*, vitamins, mineral salts, and fish oil, did not show any significant advancement in clinical periodontal parameters compared to SRP and olive oil [39]. However, this clinical study included a very small number of patients per group (*n* = 15), and this small sample size does not allow us to draw definite conclusions yet. Collagen peptides are very popular anti-inflammatory supplements, especially for the prevention of arthritis. Periodontal recall patients with a history of treated periodontitis who received collagen peptides after professional mechanical debridement showed significantly lower gingival bleeding and periodontal inflamed surface area (PISA) compared to patients who received professional mechanical debridement and placebo, decreasing the risk of re-activation of periodontitis [40]. The adjunctive intake of a nutraceutical agent containing baicalin (Neuridase^®^, Enfarma SRL, Misterbianco, Italy) with SRP in patients with moderate periodontitis significantly decreased PPD and BoP but also pain after SRP, compared to SRP and placebo [41]. More studies on nutraceutical agents and other micronutrient supplements are needed to better evaluate their adjunctive effect on periodontal therapy. 

### 2.3. Omega-3 Fatty Acids

Omega-3 (n-3) polyunsaturated fatty acid (PUFA) dietary supplementation has been promoted for chronic inflammatory disorders such as RA and cardiovascular disease [43], diabetes, cancer [44], and COVID-19. Linolenic acid (LNA) is a fatty acid that can be found in vegetables, while eicosapentaenoic acid (EPA) and docosahexaenoic acid (DHA) are derived from marine food sources. The most popular n-3 PUFA supplements include fish oil, krill oil, and algal oils. [45]. The dosing, time course, and formulations of n-3 PUFAs in clinical studies vary and remain ill-defined [10]. The anti-inflammatory effects, however, are clearly dose-dependent, and higher doses (2.0–6.0 g/day) show stronger efficacy [43,44]. n-3 PUFA supplementation decreases the systemic indicators of chronic inflammation in plasma, such as CRP, IL-6, TNF-, and adiponectin, according to recent meta-analyses of human clinical studies [45]. Markers of chronic inflammation in plasma include CRP, IL-6, TNF-α, and adiponectin [45]. Increased bleeding time, allergy to fish or ASA, diarrhea, and steatorrhea are a few possible n-3 PUFA adverse effects [10]. A longitudinal study in older Japanese patients showed that a high n-6 to n-3 PUFA diet was associated with a greater risk of periodontal disease events, highlighting the potential advantages of diets rich in omega-3 or omega-3 supplements [46]. The benefits of dietary supplementation with omega-3 (n-3) PUFAs have been described in several human clinical studies on periodontal diseases [47,48,49,50,51,52], which are described below and summarized in Table 3.

A recent systematic review that assessed omega-3 PUFAs as an adjunct to nonsurgical periodontal treatment in patients with periodontitis showed that the use of omega-3 PUFAs resulted in a significant PPD reduction and CAL gain of 0.42 mm and 0.58 mm, respectively, when compared to nonsurgical periodontal treatment alone [53]. Several studies, though, evaluate the combined effect of omega-3 with aspirin in periodontal therapy, making it harder to attribute this positive therapeutic effect only to omega-3 supplementation. In an RCT in Egypt, patients with advanced chronic periodontitis were randomized to receive SRP with either fish oil supplements (3 g) and aspirin (81 mg) or placebo capsules. A significant reduction in PPD and CAL gain was shown in the test group compared to the control at 6 months [48]. The combined daily use of omega-3 and aspirin for 2 months as an adjunct to SRP (either two months before or two months after debridement) in patients with moderate/severe periodontitis and uncontrolled diabetes mellitus type II resulted in more patients achieving the clinical endpoint of treatment (less than or equal to four sites with PPD ≥5 mm) compared to the control group. However, no significant differences in clinical periodontal outcomes between test and control groups were observed, in addition to the fact that there was no difference in the improvement in periodontal status between different administration timings of omega-3 and aspirin [47]. Low doses of omega-3 PUFAs in the diet combined with SRP caused a significant decrease in TNF-α levels in the saliva of patients with periodontitis, without improving additional clinical periodontal parameters [55]. The impact of omega-3 nutrition as a monotherapy on periodontal clinical parameters has also been investigated. When compared to placebo plus 81 mg of aspirin, the monotherapy of 2000 mg of DHA per day dramatically reduced gingival inflammation and moderate periodontitis [54]. Despite the positive indications that omega-3 PUFA supplements can be used as an adjunct to periodontal therapy, more studies with larger sample sizes and longer follow-up periods are essential to elucidate their therapeutic benefits and the most effective doses and administration.

### 2.4. Plant-Derived Compounds

Plant-derived compounds have been potential therapeutic agents recognized for their vigorous antioxidant and anti-inflammatory properties. Polyphenols are beneficial ingredients in many plant foods and can be classified into flavonoids, proanthocyanidins, phenolic acid, polyphenolic amides, and other polyphenols, such as resveratrol and curcumin [56]. The types of food that have the highest polyphenol content per serving include berries, herbs and spices, cocoa powder, flaxseeds, vegetables, olives, red wine, coffee, and tea [57]. Due to the anti-inflammatory and antibacterial actions of polyphenols and other derived nutrients, several clinical studies have investigated their contribution to the improvement in therapeutic outcomes in patients with periodontal diseases. The studies are described below and summarized in Table 4. 

A new nutritional supplement made of oligomeric proanthocyanidins (OPCs) was evaluated in an RCT using a 21-day established protocol of experimental gingivitis by quitting oral hygiene in dental students in Seville, Spain [58]. The Silness and Löe index was higher in the control group than in the test group, reaching a two-fold difference between the groups, while the gingival bleeding index was significantly lower in the test group, suggesting that this nutritional supplement made of OPCs can help in the management of gingival inflammation. A higher intake of fruits and vegetables has been associated with improved healing and therapeutic outcomes after periodontal therapy, and was discussed earlier in this review. A double-blind RCT of nonsmoking patients with periodontitis (*n* = 60) evaluated the impact of an adjunctive phytonutrient supplement after SRP [59]. The use of fruit/vegetable juice powder capsule supplementation resulted in higher plasma levels of adherence/β-carotene and an additional improvement in PPD and BOP reduction compared to placebo capsules, outcomes that can be attributed to the greater micronutrient bioavailability triggered by the supplements. In addition, the consumption of a cranberry functional beverage for 8 weeks after nonsurgical periodontal treatment in patients with gingivitis improved gingival and plaque indices without increasing any risk of caries development [26]. 

Resveratrol, a polyphenol present in various foods derived from fruits and vegetables, has been considered a promising natural drug for the prevention and therapy of various inflammatory diseases due to its ability to inhibit the oxidative damage and mitochondrial dysfunction in cells [70]. Resveratrol supplementation as a monotherapy was studied in patients with aggressive periodontitis, showing that resveratrol improved clinical periodontal outcomes (BOP, PPD, CAL, PI) by decreasing inflammatory markers and systemic endotoxin [60]. Daily supplementation with 500 mg resveratrol capsules was the recommended treatment for patients with periodontitis [60], confirming a previous study where daily supplementation with 480 mg resveratrol capsules improved PPD reduction after nonsurgical periodontal therapy in patients with periodontitis and diabetes mellitus type II [61]. More studies with larger sizes and longer follow-up periods are needed to verify these positive therapeutic benefits of resveratrol in periodontal treatment. 

Herbs have been considered a source of powerful antioxidants and polyphenols, with potential anti-inflammatory actions in the periodontium. The daily consumption of green tea had a positive adjunctive therapeutic benefit with nonsurgical periodontal therapy in patients with chronic periodontitis [62,63], attributed to an increase in the antioxidant capacity of GCF [63]. Chicory leaf extract (an herbal plant with polyphenols) capsules or placebo ones containing wheat flour were randomly administered to periodontitis patients in combination with SRP; the intervention group showed a significantly higher reduction in PPD, most likely due to increased serum levels of antioxidants [64]. The daily consumption of bilberries (blueberries) rich in antioxidants and polyphenols for one week had a positive effect on reducing BOP in patients with gingivitis by decreasing GCF levels of inflammatory markers [65].

Caffeine has been found to modulate both innate and adaptive immune responses and reduce inflammation [66]. In a large longitudinal study in males (*N* = 1152), a small but significant decrease in the number of teeth with periodontal bone loss was linked to greater coffee consumption [66]. The oral administration of propolis, a viscous substance produced by bees, has been shown beneficial in the management of periodontal inflammation. A cohort of 104 patients with gingivitis and incipient periodontitis were randomized to receive either a capsule containing propolis or placebo, without any periodontal therapy. This RCT showed that the administration of propolis reduced the gingival index and IL-6 in GCF [67]. A 6-month daily regimen of 400 mg propolis in patients with periodontitis and type II diabetes mellitus combined with SRP resulted in a significantly higher PPD reduction and CAL gain, but also in a significant reduction in HbA1C compared to the control group, who received placebo capsules [68]. Turmeric, especially its most active compound curcumin, have been found to have anti-inflammatory properties and promote wound healing by downregulating the activities of lipoxygenases and cyclooxygenases in experimental periodontitis (in vivo) [71]. However, the daily oral administration of curcumin capsules (200 mg) in periodontitis patients in Saudi Arabia after periodontal surgery (open-flap debridement for PPD elimination) did not result in any significant differences in postoperative pain and discomfort compared to mefenamic acid [69].

## 3. Conclusions

Periodontitis is a major source of oral and systemic inflammation and a significant cause for tooth loss in adults. The role of a dysregulated hyperactivated host immune response in periodontitis is clear. Differences between individuals in the response to bacterial plaque may be a result of variation in host susceptibility and immune response, with some individuals being very sensitive and developing aggressive forms of periodontitis at a relatively young age, whilst others are resistant and never develop periodontitis [4,72]. Living in the era of personalized medicine, it is of paramount importance to customize the diagnosis with risk factors for periodontal diseases and periodontal treatment for each patient.

Modifying our diet to a Mediterranean diet one that involves a high intake of vegetables, fruits, olive oil, herbs, and fatty sea fish seems to decrease gingival inflammation and bleeding on probing. A high consumption of fruit seems to counteract inflammation and favors the return to periodontal health. Although the role of vitamin D in regulating bone homeostasis has been recognized, the vitamin D supplementation had a modest effect as an adjunct to periodontal therapy, with limited clinical relevance. Limited evidence on the adjunctive effect of supplementation with other vitamins and micronutrients, such as collagen peptides, on periodontal therapy is currently available, and more clinical trials are needed. Despite the positive therapeutic effect of omega-3 PUFAs, with or without aspirin, in patients with periodontitis, more studies are needed to elucidate the most effective doses of administration of omega-3 PUFAs. Plant-derived compounds including polyphenols, herbs, and phytonutrient supplements are promising therapeutic agents in the management of periodontal inflammation; however, more clinical investigations are required to draw a decisive conclusion. In summary, several clinical studies on food ingredients and types of diet have revealed their positive impact on periodontal therapy due to their anti-inflammatory effects. The limitations of the current available clinical studies are the small sample sizes and the short follow-up periods. More human studies with larger sizes and longer follow-up periods are needed to verify these positive therapeutic benefits of food ingredients and supplements in periodontal treatment.

The nutritional assessment of our patients with questionnaires has not received adequate attention in clinical practice and needs to be more frequently integrated. It is very important that clinicians identify patients who might receive inadequate nutrients with anti-inflammatory properties in their diet and consider recommending them to modify their diet and enrich it with food compounds and supplements that may ultimately enhance the outcomes of periodontal treatment and reduce their risk of the progression of periodontal diseases. It is also critical to investigate further nutrients that would promote the maintenance of homeostasis in the periodontium, where inflammation is controlled and eubiosis (symbiosis) in the oral cavity is retained. Therefore, more clinical studies on several other food ingredients such as cocoa, honey, mastic gum, red wine, and coconut oil are needed, to investigate their effect on modifying the inflammatory response in the periodontium and improving the periodontal status and wound-healing process. Unique formulations of different nutrients in oral health products (mouthwash, toothpaste, gels, strips) need also to be explored and developed further as possible therapeutic agents for periodontal inflammation. 

## Figures and Tables

**Table 1 pathogens-12-00520-t001:** Diet and Periodontal Diseases.

Nutrient(s)	Type of Study	Methodology	Follow-Up Period	Clinical Outcomes	Subclinical Outcomes	Author(s) and Year
Mediterranean diet	RCT	42 patients with gingivitis were divided into 2 groups. Test group had to adhere to Mediterranean diet (DM) for 6 weeks and control group did not have to change their diet. Gingival parameters were assessed at baseline, week 2 (beginning of the MD intervention), and week 8.	8 weeks	Test group showed better results in gingival inflammatory parameters (GI, BOP) after treatment. No differences in dental bacterial plaque scores between test and control groups.	Test group achieved weight loss and waist compliance after treatment.	Bartha et al., 2021 [13]
Low in carbohydrates, rich in omega-3 fatty acids, vitamins C and D, antioxidants, and fiber	Pilot RCT	15 patients with gingivitis were divided into 2 groups. The test group was started on a diet low in carbohydrates, high in omega-3 fatty acids, and rich in vitamins C and D, antioxidants, and fiber for 4 weeks. Periodontal parameters were measured after 1 and 2 weeks, followed by a 2-week transition period, and then measured weekly for 4 weeks.	4 weeks	Test group showed significant improvement in GI, BOP, PI, PPD, and CAL parameters.	N/A	J. P. Woelber et al., 2016 [14]
Low in carbohydrates, rich in omega-3 fatty acids, vitamins C and D, antioxidants, and fiber	RCT	54 patients with gingivitis were divided into 2 groups. The test group was started on a diet low in carbohydrates, high in omega-3 fatty acids, and rich in vitamins C and D, antioxidants, and fiber for 4 weeks. Periodontal parameters were measured once a week for 4 weeks, followed by a 2-week transition period, and then measured weekly for 4 weeks.	4 weeks	Test group showed significant improvement in BOP and GI.	N/A	Sava Sunari Rajaram et al., 2021 [15]
High-fiber and low-fat diet	Pilot Study	47 volunteers were included in the study. Subjects received a high-fiber, low-fat test meal 3x/day for 8 weeks, followed by a regular diet for 24 weeks. Periodontal parameters were evaluated at the beginning and end of treatment.	8 weeks	The high-fiber, low-fat diet effectively improved PPD, CAL, and BoP in patients after treatment.	After treatment, there was improvement in metabolic profiles (body weight, HbA1c, and high-sensitivity C-reactive protein levels).	Keiko Kondo et al., 2014 [16]
Nordic diet	Cohort	2187 healthy Finns eating a Nordic diet were divided into two age groups and then into two oral hygiene groups (good and poor oral hygiene). Periodontal parameters were used as outcome variables. Dietary data were collected using a validated food frequency questionnaire.	6 weeks	Nordic diet provides evidence that it is associated with less gingival bleeding and reduced PPD in patients with poor oral hygiene.	N/A	Jauhiainen L et al., 2016 [17]
Nitrate	RCT	44 patients with gingivitis were divided into 2 groups. Test group received 100 mL of a lettuce juice drink (200 mg of nitrate) to be consumed daily for 14 days, and the control group received placebo. Periodontal parameters of salivary nitrate were evaluated before and after treatment.	2 weeks	Test group showed better results in GI on day 14.	Test group showed higher levels of salivary nitrate.	Jockel-Schneider Y et al., 2016 [18]
Nitrate	RCT	37 patients with gingivitis and reduced periodontium were divided into 2 groups. Test group received lettuce juice (200 mg of nitrate) daily for 14 days and test group received placebo. Microbial samples, salt collection, and assessment of gingival inflammation were analyzed before and after treatment.	2 weeks	Test group showed reduction in gingival inflammation after treatment.	Test group showed compositional changes within the subgingival microbiome after treatment.	Yvonne Jockel-Schneider et al., 2020 [19]
Atkins low-carb ketogenic diet	Pilot Clinical Trial	20 patients were placed on a ketogenic diet. Adherence was monitored by measuring their urinary ketones daily and keeping 7-day records. Periodontal, physical, and serological parameters were evaluated at baseline and after treatment.	6 weeks	No changes in clinical periodontal parameters. Tendency to lower plaque values after treatment.	Reduction in body weight and BMI after treatment.	Johan Peter Woelber et al., 2021 [20]
Fruit	RCT	30 patients were included in the study; they were randomized to supplement their diet with extra fruits or nuts, each at +7 kcal/for 2 months. Periodontal parameters were analyzed before and after treatment, as well as serum vitamin C, β-carotene/cholesterol, and α-tocopherol/cholesterol.	8 weeks	Significant reduction in the number of subjects with PPD ≥4 mm in group with fruits compared to nuts, while PI and BoP remained unchanged in both groups.	Vitamin C levels increased in both groups and α-tocopherol/cholesterol increased in the fruit group and decreased in the nut group, as did β-carotene/cholesterol.	Sara Fridell et al., 2018 [21]
Kiwifruit	RCT	50 patients were divided into 2 groups. Test group received 2 kiwis/day for 5 months and control group did not. The first SRP was performed after 2 months of treatment. Periodontal parameters and blood samples were evaluated after 2 and 5 months.	2 and 5 months	Test group showed reduced gingival bleeding in 2 months, which was also maintained after treatment.	Systemic biomarkers and vital signs showed no clinically relevant differences between groups.	Filippo Graziani et al., 2017 [22]

Abbreviations: RCT: Randomized controlled trial; PPD: Probing pocket depth; CAL: Clinical attachment level; SRP: Scaling and root planing; GCF: Gingival crevicular fluid; BOP: Bleeding on probing; GI: Gingival index; PI: Plaque index; BMI: Bone Mass Index; N/A: Not available.

**Table 2 pathogens-12-00520-t002:** Vitamins and Periodontal Diseases.

Nutrient(s)	Type of Study	Methodology	Follow-Up Period	Clinical Outcomes	Subclinical Outcomes	Author(s) and Year
Vitamin D	RCT	360 patients with moderate or severe periodontitis received nonsurgical periodontal treatment, and after 3 months were divided into 3 groups: 2000 IU/day of vitamin D3, 1000 IU/day of vitamin D3, and placebo. Periodontal parameters were evaluated at the beginning and end of treatment.	3 months	Vitamin D supplementation in both dose groups resulted in slight but significant reduction in PPD and CAL for moderate and deep periodontal pockets.	N/A	Weimin Gao et al., 2019 [29]
Vitamin D	RCT	37 periodontitis patients with serum vitamin D3 levels below 30 ng/mL were divided into 2 groups. Both groups received initial SRP treatment. Test group received 25,000 IU/week of vitamin D3 and control group received placebo for 6 months. Periodontal parameters were evaluated at 1, 3, and 6 months of treatment.	6 months	Test group showed better results in PPD reduction after treatment.	N/A	Marina Peri’c et al., 2020 [30]
Vitamin D	RCT	40 patients with periodontitis were divided into 2 groups. Both groups received SRP treatment. Test group received 400 IU/day for 6 weeks and control group did not. Periodontal parameters and serum vitamin D levels were evaluated before and after periodontal treatment, and after 6 weeks of vitamin D treatment.	6 weeks	Intragroup comparison of clinical parameters from baseline to 6 weeks showed a statistically significant reduction in both groups. No differences observed between comparison groups.	Bone mineral density was evaluated by a qualitative ultrasound; however, no significant difference was observed between comparison groups.	Shree Mohan Mishra et al., 2022 [31]
Vitamin D	RCT	19 patients with GAgP periodontitis received initial periodontal treatment. CGF collection was performed before therapy and after 2 and 6 months. Plasma collected before therapy and after 2 months to compare systemic and local levels of 25-hydroxyvitamin D3, osteocalcin, and interleukin-1b and -6 before and after treatment.	6 weeks	GCF levels of vitamin D3 and IL-1β decreased significantly from baseline to 2 and 6 months after therapy.	Systemic levels of vitamin D3 and IL-1β were reduced significantly from baseline to 2 months after therapy.	Kaining Liu et al., 2010 [32]
Vitamin D	Pilot RCT	23 patients with dark skin were randomized into 2 groups. Test group received 4000 IU/day of vitamin D and control group received placebo for 8 weeks, after which time both groups received SRP and continued to use their supplements for another 8 weeks. At 8 and 16 weeks, saliva was collected to assess the presence of inflammatory cytokines.	16 weeks	Test group showed reduction in cytotoxic T lymphocytes (CD3 and CD8) in the blood and reduced salivary cytokines, but increased proteins related to autophagy.	N/A	Mohamed M. Meghil et al., 2019 [33]
Vitamin C	RCT	30 patients with periodontitis were divided into 2 groups. Both groups received SRP treatment. Test group received vitamin C. Periodontal parameters and plasma antioxidant capacity (TAOC) levels were evaluated at the beginning and end of treatment.	4 weeks	There was no effect on clinical periodontal parameters after treatment; only 20% of the periodontitis patients were vitamin-C- depleted (plasma concentration < 4 mg/L).	TAOC levels were lower in patients with periodontitis after treatment.	Ali E. Abou Sulaiman et al., 2010 [34]
Vitamin C and Omega-3	Case-Control	Periodontitis patients: total number *N* = 373 were further divided into: 245 periodontitis without tooth loss (POL) and 128 periodontitis with tooth loss (PWL). They were matched to 373 controls. Food Frequency Questionnaire was used to collect data on vitamin and omega-3 fatty acid intake, as well as their dental status. Blood samples were collected for metabolite concentrations analysis.	Not specified	Not specified	Higher intake of vitamin C in periodontitis group (*p* = 0.007) compared to control. Periodontitis with tooth loss showed a significantly elevated daily vitamin C intake compared to control (*p* = 0.02). No significant difference between the groups in the total uptake of omega-3 fatty acids.	Mewes et al., 2022 [35]
Vitamin E	RCT	38 patients with periodontitis were divided into 2 groups. Both groups received SRP treatment. Test group received 200 mg of vitamin E every other day for 3 months. Periodontal parameters and salivary superoxide dismutase (SOD) activity were evaluated before and after treatment.	3 months	Test group showed better results in all analyzed periodontal parameters (PPD, BOP, PI, GI, CAL) after treatment.	Test group showed better results in salivary superoxide dismutase (SOD) activity after treatment.	Neha Singh et al., 2013 [36]
Vitamin C, E, lysozyme and carbazochrome	RCT	100 patients were divided into 2 groups. Test group received a fixed dose of the combination of vitamin C, vitamin E, lysozyme, and carbazochrome (CELC) combined with SRP, and control group received SRP and placebo for 4 weeks. Both groups received vitamins for another 4 weeks. Periodontal parameters were evaluated at 4 and 8 weeks.	8 weeks	Test group showed GI reduction after 4 and 8 weeks, but without any differences in other periodontal parameters between studied groups.	N/A	Ji-Youn Hong et al., 2019 [37]
Folic Acid	RCT	60 patients with periodontitis were divided into 2 groups. Both groups received initial SRP. Test group received folic acid and control group received placebo 3x/day for 4 weeks. Periodontal parameters and crevicular fluid were assessed at baseline and after 1, 3, and 6 months.	4 weeks	Significant time-dependent reduction was detected in all clinical parameters for both groups. Test group showed better results in CAL at months 1 and 3, and in GI at month 1.	Test group showed more homocysteine (Hcy) at months 3 and 6. GCF and Hcy volume showed reduction after treatment in both groups.	Keceli et al., 2020 [38]
Micronutrient complex	RCT	30 patients with periodontitis adhered to the Mediterranean diet and SRP treatment and took a micronutrient complex or olive oil twice a day for 3 months. Periodontal parameters were correlated with serum C-reactive protein and MMP-8/9 salivary matrix, quantified at 1 and 3 months of treatment.	3 months	Both groups showed better periodontal parameters at all time points. No differences between studied groups.	SRP and micronutrient complex resulted in reduction in salivary MMP-8/-9 at T2, and of MMP-9 at T1, while SRP and olive oil did not induce any significant changes in MMP-8/-9.	Giulio Rasperini et al., 2019 [39]
Collagen peptide	RCT	39 periodontal recall patients with history of treated periodontitis were divided into 2 groups. Test group received sachets containing a specific preparation of collagen peptide and control group received placebo 1 time/day for 90 days. Both groups received professional mechanical debridement.	3 months	The addition of collagen peptide showed better results in BoP, PISA, and GI after treatment compared to placebo after professional mechanical debridement.	N/A	Yvonne Jockel-Schneider et al., 2022 [40]
Nutraceutical agent (Neuridase^®^, Enfarma SRL, Misterbianco, Italy)	RCT	66 patients with moderate periodontitis were divided into 2 groups. Test group received nutraceutical agent and SPR while control group received SRP and placebo.	6 months	Test group showed better results in periodontal parameters (PPD, BOP, CAL) after 30 and 60 days compared to control group.	Test group showed greater reduction in inflammatory mediators, and change in pain (VAS) 6, 12, 24, and 48 h after SRP.	Gaetano Isola et al., 2021 [41]

Abbreviations: RCT: Randomized controlled trial; PPD: Probing pocket depth; CAL: Clinical attachment level; SRP: Scaling and root planing; GCF: Gingival crevicular fluid; BOP: Bleeding on probing; GI: Gingival index; PI: Plaque index; PISA: Periodontal inflamed surface area; VAS: Visual analogue scale; MMP: Matrix metalloproteinase; IL: interleukin; N/A: Not available.

**Table 3 pathogens-12-00520-t003:** Omega-3 Fatty Acids and Periodontal Diseases.

Nutrient(s)	Type of Study	Methodology	Follow-Up Period	Clinical Outcomes	Subclinical Outcomes	Author(s) and Year
Dietary ratio of n-6 to n-3 polyunsaturated fatty acids	Cohort	Dietary intake assessment of 235 eligible older Japanese subjects.	3 years	A high dietary n-6 to n-3 PUFA ratio was significantly associated with a greater risk of periodontal disease events (number of teeth with periodontal disease progression for three years).	N/A	Masanori Iwasaki et al., 2011 [46]
Omega-3	Systematic Review/ Meta-Analysis	Assessed only RCTs with minimum 3-month follow up of SRP with and without omega-3 supplements.	3–6 months	Additional PPD reduction and higher CAL gain in patients who received omega-3 fatty acid dietary supplementation with SRP compared to SRP alone.	N/A	Nidia C. Castro dos Santos et al., 2022 [53]
Omega-3 and Aspirin	RCT	Subjects with advanced chronic periodontitis (*N* = 80): The test group received SRP, 3 g of fish oil, and 81 mg of aspirin, while the control group received SRP and placebo capsules.	6 months	Additional PPD reduction, higher CAL gain, and higher reduction in number of sites with PPD ≥5 mm for test group compared to control.	Significant reduction in salivary MMP-8 and RANKL levels in test group compared to the control.	El-Sharkawy et al., 2010 [48]
Omega-3 and Aspirin	RCT	Subjects with moderate periodontitis (*N* = 46): The test group received 2 g DHA and 81 mg aspirin and the control group received placebo capsules and 81 mg aspirin. No periodontal therapy.	3 months	Additional PPD reduction of 0.17 mm and higher reduction in number of sites with PPD ≥5 mm for DHA and ASA.	Significant reduction in CRP and IL-1β in GCF but not IL-6 for DHA and aspirin.	Naqvi et al., 2014 [54]
Omega-3 and Aspirin	RCT	Patients with moderate to severe chronic periodontitis with grade II furcation (*N* = 40). Both groups received open-flap debridement with bone graft (DFDBA). The test group received omega-3 (3 g fish oil) +75 mg aspirin for 6 months, while the control group received placebo capsules.	6 months	Additional PPD reduction of 0.7 mm and CAL gain of 0.4 mm for omega-3 and aspirin group.	Significant reduction in IL-1β in GCF for omega-3 and aspirin group.	ElKhouli AM et al., 2011 [49]
Omega-3 and Aspirin	RCT	Chronic periodontitis patients with diabetes mellitus type II (*N* = 40). The test group received omega-3 (3 g) + aspirin (75 mg) following SRP for 6 months.	6 months	Significant reduction in PD, CAL, and GI after 3 months and 6 months in the test group compared to control.	Significant reduction in GCF levels of MCP-3 and IL-1β at 3 and 6 months in the test group compared to control.	Elwakeel et al., 2015 [50]
Omega-3 and Aspirin	RCT	Patients with type II DM (uncontrolled) and generalized moderate/severe (stage III and IV; grade B and C) periodontitis patients were divided into 3 groups (*N* = 25/group): control group (CG): placebo; test group 1 (TG1): 3 g of fish oil + 100 mg ASA daily for 2 months after periodontal debridement; test group 2 (TG2): 3 g of fish oil + 100 mg ASA daily for two months before periodontal debridement.	6 months	A higher number of patients in both test groups (TG1, TG2) achieved the clinical endpoint of treatment (less than or equal to four sites with PPD ≥5 mm) compared to control group. No differences in timing of administration of omega-3 and aspirin. No differences in clinical periodontal parameters among groups.	Significant reduction in HbA1c only in TG1 compared to TG2 and CG. Significant reduction in GCF levels of IFN-*γ* and IL-8 in both test groups, while IL-6 GCF levels were lower only for TG1.	Nidia C. Castro dos Santos et al., 2020 [47]
Omega-3	RCT	Generalized severe periodontitis subjects (stage III and IV) (*N* = 30). Test group (*n* = 16) received SRP and fish oil for 3 months twice a day. Control group (*n* = 14) received SRP alone.	3 months	Significant BOP reduction, higher CAL gain, and higher number of sites with closed pockets (PPD ≤ 4 mm) for test group compared to control.	Significantly higher salivary levels of IL-10 and markedly lower levels of IL-8 and IL-17 in test group compared to control.	Mirella Stańdo et al., 2020 [52]
Omega-3	RCT	Chronic periodontitis patients allocated into two groups with equal number of participants (*N* = 15) and received SRP. The test group was supplemented with low-dose Omega-3 PUFAs 6.25 mg EPA and 19.19 mg docosahexaenoic acid.	6 months	No additional clinical benefit for PPD reduction and CAL gain for low dose n-3 PUFAs.	Significant reduction in salivary TNF-α for low-dose n-3 PUFAs.	Keskiner I et al., 2017 [55]
Omega-3	RCT	90 patients with periodontitis. Test group received SRP and omega-3 supplements 500 mg BD daily for 1 month. Control group received only SRP.	3 months	Significant reduction in PPD and CAL gain in test group compared to control. Significant reduction in GI in test group, but no differences in PI between groups.	N/A	Shirish K. Kujur et al., 2020 [51]

Abbreviations: RCT: Randomized controlled trial; PPD: Probing pocket depth; CAL: Clinical attachment level; SRP: Scaling and root planing; EPA: Eicosapentaenoic acid; DHA: Docosahexaenoic acid; ASA: Aspirin; PUFAs: Polyunsaturated fatty acids; GCF: Gingival crevicular fluid; BOP: Bleeding on probing; GI: Gingival index; PI: Plaque index; MMP: Matrix metalloproteinase; IL: Interleukin; TNF: Tumor necrosis factor; N/A: Not available.

**Table 4 pathogens-12-00520-t004:** Plant-Derived Compounds and Periodontal Diseases.

Nutrient(s)	Type of Study	Methodology	Follow-Up Period	Clinical Outcomes	Subclinical Outcomes	Author(s) and Year
Oligomeric proanthocyanidin nutritional supplement	RCT	*N* = 20 dental students followed a 21-day protocol of experimental gingivitis and randomly either received oligomeric proanthocyanidin nutritional supplements treatment consisting of 90 mg exocian cran 408 and 120 mg of vitamin C (test) or placebo capsules (control). Oral hygiene was not performed for 21 days.	21 days	Silness and Löe index and gingival bleeding index were significantly lower in the test group compared to the control. Plaque index was significantly higher in the test group.	GCF levels of IL-6 were significantly lower in the test group (22.15 ± 15.14 pg/mL) compared to the control one (69.40 ± 50.10 pg/mL) (*p* = 0.011).	R.M Diaz Sanchez et al., 2017 [58]
Encapsulated fruit, vegetable, and berry juice	RCT	A total of 60 nonsmoking subjects with periodontitis were randomized into 3 groups: Daily supplementation with capsules with fruit/vegetable (FV) juice powder, fruit/vegetable/berry (FVB) juice powder, and placebo (control) for 2 months after SRP.	8 months	PPD, CAL, and BOP at 2 months were improved in all groups, with additional improvement in the FV group compared to the placebo one (*p* < 0.03). FV groups showed better BOP % and plaque scores when compared to control at 5 months (*p* < 0.05). No differences between FV and FVB groups.	Adherence/β-carotene plasma levels were significantly higher in both FV and FVB groups compared to control (*p* < 0.001). GCF volume was significantly reduced in both supplement groups (FV, FVB), compared to control.	Chapple et al., 2012 [59]
Cranberry functional beverage	RCT	50 gingivitis patients were randomized into two groups: The test group received a daily 750 mL of cranberry functional beverage (CFB) for 8 weeks, while the control group was given the same amount of water, both combined with nonsurgical periodontal therapy.	8 Weeks	Significant reduction in GI and PI scores in the test group compared to control, but no differences in BOP scores between the two groups.	Saliva and serum levels of total anti-oxidant status, malonyldialdehyde, and IL-1β were not significantly different between comparison groups. Number of *Streptococcus mutans* were reduced in the test group but not in the control.	Wozniewicz et al., 2018 [26]
Resveratrol	RCT	Patients with aggressive periodontitis (*N* = 160) were randomized into 4 groups: high dose of resveratrol (RV) (RV 500 mg/d), middle-dose (RV 250 mg/day), low-dose (RV 125mg/d), and placebo capsules (oral administration). No periodontal therapy was provided.	8 weeks	Significant improvement was observed in the CAL, BI, OHI-S, and PPD in test groups compared to placebo group (*p* < 0.01). High-dose and middle-dose RV groups showed significant differences compared to low-dose RV group. However, the difference between high-dose and middle-dose RV groups was not statistically significant (*p* > 0.05).	RV supplementation reduced inflammatory markers and endotoxin in serum and GCF compared to placebo capsules. No difference was found between different RV doses.	Qiang, Zhang et al., 2021 [60]
Resveratrol	RCT	Patients with periodontitis and diabetes mellitus type II (*N* = 43) were randomized into receiving either resveratrol capsules (480 mg of resveratrol) or placebo capsules daily. Nonsurgical periodontal treatment was also provided to all patients.	4 weeks	PPD were significantly lower in the test compared to the control groups after intervention (2.35 ± 0.6 mm and 3.38 ± 0.5 mm, respectively).	Mean serum levels of fasting insulin and insulin resistance: lower in the test group compared to the control (*p* < 0.05). No difference in serum fasting blood glucose levels and triglycerides between two groups.	Zare Javid et al., 2017 [61]
Green Tea	RCT	Chronic periodontitis patients (*N* = 30) were randomized into two groups; all participants received SRP. The test group consumed green herbal tea while the control group did not consume anything.	6 weeks	Greater reduction in PPD and BI in the test group compared to the control. PI difference was not significant among the two groups.	N/A	F.Taleghani et al., 2018 [62]
Green tea	RCT	Mild to moderate chronic periodontitis patients (*N* = 120) received SRP and were randomized to consume either green tea sachets (test) or placebo cellulose sachets (control). Patients were asked to drink two cups per day.	3 months	GI, PI, BoP, and PPD reduction was significantly lower in the test group compared to the control. CAL gain was significantly higher in the test group (2.01 + 0.65 mm) compared to control (1.60 + 0.54 mm) (*p* < 0.001).	GCF antioxidant levels significantly increased in the test group (*p* < 0.001)	Chopra et al., 2016 [63]
Chicory leaf extract capsules	RCT	Chronic periodontitis patients (*N* = 40) received SRP and were randomized to receive either chicory leaf capsules (2 g) or placebo capsules (containing 1 g wheat flour) daily.	8 weeks	PPD showed a significant reduction in test group compared to control group.	Total antioxidant capacity (TAC) and uric acid levels in plasma increased significantly in the intervention group compared to control, while lipid (LDL-C, HDL-C, TC, TG) levels decreased.	Babaei et al., 2018 [64]
Bilberries	RCT	(*N* = 24) subjects with gingivitis were divided into placebo and two test groups who either consumed 250 g or 500 g of bilberries daily.	1 week	The mean reduction in BOP before and after intervention was 31% in the placebo group, 41% for those who consumed 250 g, and 59% for 500 g of bilberries/day.	Significant reduction in IL-1β, IL-6, and VEGF in GCF samples in the test group that consumed 500 g bilberries.	Cecilia Widén et al., 2015 [65]
Coffee consumption	Cohort	*N* = 1152 males, periodontal status recorded, including radiographical bone loss scores, PPD, BOP, calculus, and plaque, and self-reported coffee intake assessments (from 1968 up to 1998).	30 years	Moderate-to-severe alveolar bone loss decreased as coffee consumption increased.	N/A	Nathan Ng et al., 2014 [66]
Propolis	RTC	104 patients with gingivitis and incipient periodontitis were randomized into 2 groups: Test group ingested a capsule containing propolis daily for 8 weeks, while the control group received a placebo capsule. SRP was provided to all patients at the end of the study (8 weeks).	8 weeks	Test group showed significant improvement in gingival index after 4 and 8 weeks of treatment compared to control group.	In the test group, IL-6 was reduced and MMP-9 increased after 8 weeks.	Jin-Young Park et al., 2021 [67]
Propolis	RTC	50 patients with periodontitis and type 2 diabetes mellitus received initial SRP and were divided into 2 groups. Test group received 400 mg of propolis orally 1x/day for 6 months, while the control group received placebo capsules.	3 and 6 months	Test group showed significantly higher PPD reduction and CAL gain after 3 and 6 months of treatment.	Test group showed significant reduction in HbA1C after treatment.	El-Sharkawy et al., 2016 [68]
Curcumin (turmeric)	RTC	76 patients with moderate/severe periodontitis were randomized into 2 groups: Test group received curcumin capsules (200 mg) and control group received 400 mg of mefenamic acid after surgical periodontal therapy (open-flap debridement). All patients received antibiotic treatment as well. Patients’ pain was assessed using the numerical rating scale and verbal rating scale after 24, 48, and 72 h.	7 days	Curcumin did not result in any significant differences in postoperative pain and discomfort in patients compared to mefenamic acid.	N/A	Mansour Al Askar et al., 2022 [69]

Abbreviations: RCT: Randomized controlled trial; PPD: Probing pocket depth; CAL: Clinical attachment level; SRP: Scaling and root planing; OHI-S: Oral hygiene index—simplified; GCF: Gingival crevicular fluid; BOP: Bleeding on probing; GI: Gingival index; PI: Plaque index; RV: Resveratrol; IL: Interleukin; TC: Total cholesterol; TG: Triglycerides; LDL-C: Low-density lipoprotein cholesterol; HDL-C: High-density lipoprotein cholesterol; VEGF: Vascular endothelial growth factor; N/A: Not available.

## Data Availability

Not applicable.

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
