# Peer review of "Anti-Inflammatory Benefits of Food Ingredients in Periodontal Diseases"

_pathogens, 2023, doi:10.3390/pathogens12040520_

Round 1
Reviewer 1 Report
Introduction
The pathophysiology of the disease is explained very well. However, the need of the study has to be better established. The introduction should be focused on the food and nutritional supplements and their potential role in the patho-physiology of Periodontitis. Are there any similar reviews conducted previously? If yes, how does this review add to the existing information?
Methods
The search strategy needs to be better explained. Why only one database was included for the search? Line no 79-81 has to be revised as it is too long and has grammatical errors. The final number of studies included in the review has not been mentioned.
Results
Małgorzata Woźniewicz et. al, 2018- this study has been included twice under the sub-heading Diet and Plant-derived Compounds. Kindly clarify.
Conclusion
Kindly add the limitations of the study. The conclusions are lack-lusture and just a general comment on the importance of nutritional supplements. Please add on the research gap identified to guide future researchers. Kindly highlight the more promising nutritional agents identified by your review.
Author Response
- Introduction: The pathophysiology of the disease is explained very well. However, the need of the study has to be better established. The introduction should be focused on the food and nutritional supplements and their potential role in the patho-physiology of Periodontitis. Are there any similar reviews conducted previously? If yes, how does this review add to the existing information?
Reply: We would like to thank reviewer 1 for the positive comments. We added some more information at the end of the introduction to highlight the distinction of this submitted review compared to previous reviews in the field.
- Methods. The search strategy needs to be better explained. Why only one database was included for the search? Line no 79-81 has to be revised as it is too long and has grammatical errors. The final number of studies included in the review has not been mentioned.
Reply: We did use another search engine (Web of Science) that confirmed our previous findings. We added more information and details about the studies that we found initially and how many clinical studies were finally used and presented in tables after filtering further the available studies.
- Results: Małgorzata Woźniewicz et. al, 2018- this study has been included twice under the sub-heading Diet and Plant-derived Compounds. Kindly clarify.
Reply: We would like to thank reviewer 1 for bringing this point to our attention. The study by Małgorzata Woźniewicz et. al, 2018 was removed from table 1 (diet) and is now included only in table 4 (plant-derived compounds).
- Conclusions: Kindly add the limitations of the study. The conclusions are lack-lusture and just a general comment on the importance of nutritional supplements. Please add on the research gap identified to guide future researchers. Kindly highlight the more promising nutritional agents identified by your review.
Reply: We would to thank reviewer 1 for the suggestion. We added more information as suggested in the “Conclusions” about the most promising nutritional agents and the limitations highlighting what researchers need to conduct in the future.
Reviewer 2 Report
This work presents a critical review of recent evidence on the activity (anti-inflammatory effect) of food ingredients and supplements on periodontal disease based on human clinical studies. The authors analyzed and classified different food ingredients and supplements and nicely presented these groups in appropriate tables giving a very good overview of their activity. They analyzed as groups: different types of diets (group 1), vitamins (group 2), omega-3 fatty acids (group 3) and plant derived compounds (group 4) and presented discussion for each of them. In conclusion they stated that several clinical studies have shown a positive anti-inflammatory effect on periodontal therapy. They also proposed more clinical studies with larger groups and longer study and follow up periods and some other food ingredients as possible candidates for investigation in order to improve the periodontal status and therapy.
This work represents a significant contribution to the field with a good overview what has been achieved until now.
The work is well organized, comprehensively described and easy to read and follow.
The work is scientifically sound and not misleading.
The cited references are appropriate, relevant and sufficient.
The manuscript is well written and the English is correct and readable but needs spell checking, some corrections in tables: ß-carotene
Author Response
1.This work presents a critical review of recent evidence on the activity (anti-inflammatory effect) of food ingredients and supplements on periodontal disease based on human clinical studies. The authors analyzed and classified different food ingredients and supplements and nicely presented these groups in appropriate tables giving a very good overview of their activity. They analyzed as groups: different types of diets (group 1), vitamins (group 2), omega-3 fatty acids (group 3) and plant derived compounds (group 4) and presented discussion for each of them. In conclusion they stated that several clinical studies have shown a positive anti-inflammatory effect on periodontal therapy. They also proposed more clinical studies with larger groups and longer study and follow up periods and some other food ingredients as possible candidates for investigation in order to improve the periodontal status and therapy.
This work represents a significant contribution to the field with a good overview what has been achieved until now.
2.The work is well organized, comprehensively described and easy to read and follow.
3.The work is scientifically sound and not misleading.
4.The cited references are appropriate, relevant and sufficient.
5 The manuscript is well written and the English is correct and readable but needs spell checking, some corrections in tables: ß-carotene.
Reply: We would like to thank reviewer 2 for the positive comments. We made correction in both table 4 and in the manuscript text for β-carotene as suggested.
Reviewer 3 Report
Manuscript of considerable interest for the dental sector, before being published it needs a major revision.
Abstract to highlight the results obtained, in order to decipher the results obtained at first glance,
Keywords: they are few, insert them for further specifications
Introduction: add all minimally invasive systems for oral microbiota maintenance and all applications of all natural substances for periodontal health maintenance, and what are the pros and cons compared to the gold standard. as already studied by the research group of
Prof Scribante, probiotics, paraprobiotics, postbiotics
The new classification of periodontal disease is missing
Results: highlight the data found. Rearranging tables, very confusing
Discussion: add the use of natural substances to reduce the incidence of dysbiosis
Materials and methods: the consort flow chart is missing
Conclusions add proactive action for the maintenance of constant eubiosis for both hard and soft tissues
Bibliography:
add bibliography required
Author Response
1.Manuscript of considerable interest for the dental sector, before being published it needs a major revision.
Reply: We would like to thank reviewer 3 for the positive comments.
2.Abstract to highlight the results obtained, in order to decipher the results obtained at first glance.
Reply: We added key results in the abstract as suggested.
3.Keywords: they are few, insert them for further specifications
Reply: Several keywords were present in our first draft of our manuscript.
4.Introduction: add all minimally invasive systems for oral microbiota maintenance and all applications of all natural substances for periodontal health maintenance, and what are the pros and cons compared to the gold standard. as already studied by the research group of Prof Scribante, probiotics, paraprobiotics, postbiotics.
Reply: Mainly these probiotics, paraprobiotics, postbiotics have been included in oral formulations (toothpaste, mouthwash) and have to be excluded from this review as we focus on the anti-inflammatory effects of oral intake of food supplements.
5.The new classification of periodontal disease is missing
Reply: We used the same classification system as it was originally described in the paper/study that we cite and refer to. We have both used the classification of periodontal diseases before the AAP/EFP World Workshop (2017/2018) and the new classification accordingly.
6.Results: highlight the data found. Rearranging tables, very confusing
Reply: We would appreciate if the reviewer could be more specific on the requested modifications. The other reviewers found our tables very informative. We also believe that the results are highlighted in the manuscript text and presented in more details in the tables that display the design and results of each clinical study in an organized structured manner.
7.Discussion: add the use of natural substances to reduce the incidence of dysbiosis
Reply: As we responded to a previous comment from reviewer 3, we focus here on the anti-inflammatory effects of oral intake of food supplements and not any antibacterial properties that could be an interesting topic of another review paper. The only natural substances that reduce the incidence of dysbiosis could be the probiotics but they cannot be included in our review. Thank you for your suggestion.
8.Materials and methods: the consort flow chart is missing.
Reply: We added more details in “Methods” about our search strategy and the final number of studies used, describing our approach filtering the available eligible studies. Please note that this manuscript is a review paper and not a systematic review/meta-analysis. Thank you.
9.Conclusions add proactive action for the maintenance of constant eubiosis for both hard and soft tissues
Reply: We add a statement for the maintenance of eubiosis in “Conclusions” as suggested.
10.Bibliography: add bibliography required
Reply: “Bibliography” section has replaced the previous section “References” as suggested.
Round 2
Reviewer 3 Report
The manuscript has been revised correctly, it is only missing as already mentioned in the first round of the revision, if you do not want to include them in the introduction, you can discuss the use of probiotics, postbiotics and paraprobiotics, such as food derivatives as already studied by the group of research of Prof Scribante
Author Response
Comment:
The manuscript has been revised correctly, it is only missing as already mentioned in the first round of the revision, if you do not want to include them in the introduction, you can discuss the use of probiotics, postbiotics and paraprobiotics, such as food derivatives as already studied by the group of research of Prof Scribante
Reply: Thank you very much for your additional comment. We tried to adapt your recommendation but the use of probiotics/postbiotics and paraprobiotics by Prof. Scribante do not seem to fit in the discussion or in any other part of the manuscript.
Mainly these probiotics, paraprobiotics, postbiotics have been included in oral formulations (toothpaste, mouthwash) and have to be excluded from this review as we focus on the anti-inflammatory effects of oral intake of food supplements.
If there is any specific reference or any specific section that needs to be added, please let us know. Thank you.